# Compound Brands and the Multi-Creation of Brand Associations: Evidence from Airports and Shopping Malls

Isaac Levi Henderson [1,*], Mark Avis [2], Wai Hong Kan Tsui [3], Thanh Ngo [1] and Andrew Gilbey [1]

1   School of Aviation, Massey University, 47 Airport Drive, Palmerston North 4414, New Zealand
2   School of Communication, Journalism and Marketing, Massey University, Tennent Drive, Palmerston North 4410, New Zealand
3   School of Business, University of Southern Queensland, Toowoomba, QLD 4350, Australia
*   Correspondence: i.l.henderson@massey.ac.nz; Tel.: +64-6-951-9432

**Abstract:** The authors identify a new type of brand concept, which they term as a compound brand. Compound brands have their brand associations multi-created such that the focal brand entity, their tenants, and ancillary entities all act as sources of primary brand associations. To test the possibility of compound brands, two potential compound brands are studied, airports and shopping malls. This was completed by undertaking 480 semi-structured interviews (240 for each entity) to identify the underlying brand association structure and which associations are important for consumer brand choice. Thematic analysis was used to analyse the qualitative data. Participant responses support that compound brand association structures are created by the focal branded entity (e.g., an airport), its tenants (e.g., shops and restaurants), as well as ancillary entities (e.g., location and customers). The contributions of tenants and ancillary entities towards the brand association structures of airports and shopping malls were also statistically significant with large effect sizes. A continuum exists as to how much of the compound brand's association structure is created by its tenants, with statistically significant differences between airports and shopping malls in terms of how much tenants contribute to overall brand association structures for the compound brand.

**Keywords:** compound brands; brand associations; consumer psychology; consumer experience; marketing management; airports; shopping malls

## 1. Introduction

Sometimes research can produce novel findings as a result of serendipity [1,2]. This paper presents a new marketing concept underpinned by empirical research. It began as an investigation into the creation of airport brand associations, on the premise that airports may have their brand associations created differently due to the presence of multiple actors within an airport space (e.g., airlines, security, shops, restaurants, etc.). Our initial investigation (presented as part of the results in this study) showed that, as suspected, other actors played an important part in both creating airport brand associations, and in determining airport brand choice. It became clear that no existing brand type in the literature addressed the peculiarities of airport brand association structures. Accordingly, our initial study showed clear evidence that airport brand associations were created differently from other types of brands—which had not yet been identified in the extant literature. However, rather than focussing only on the implications for airports, it was identified that the characteristics that made an airport brand unique may also apply to other types of entities (e.g., casinos, theme parks, etc.), and thus other types of entities may also have their brand associations created in a unique way. Specifically, the presence of tenants and ancillary entities within the branded entity's "owned" space meant that consumers compounded the brand associations of the focal branded entity (e.g., an airport) with its tenants (e.g., airlines, food providers), and other entities (e.g., government security) such that the associations with

those tenants and other entities also became associations with the branded entity itself. This is conceptualised by the authors as a *compound brand*.

In considering the unique characteristics of compound brands, shopping malls seemed to be another likely candidate because their tenants and other entities also play a role in creating or hindering value creation within their "owned" space [3]. Accordingly, the investigation on airports was replicated for shopping malls, and both airports and shopping malls were treated as case studies. The results for shopping malls confirmed that the "compound" nature of their brands was even more pronounced than for airports. This led to the conclusion that a compound brand continuum exists whereby different types of entities will be positioned differently according to how important tenants and ancillary entities are in the creation of compound brand associations and in brand choice.

A key foundation for brand research has been the examination of associations that are linked via memory to a brand name [4–6]. Strong, unique, and favourable brand associations are proposed to be the source of customer-based brand equity because they affect how consumers respond to the marketing of the brand [7]. In addition, brand associations have been linked to behavioural brand loyalty [8], brand preference [9], consumer response [10], brand extension evaluation [11,12], as well as having influence on other fields of brand performance (e.g., brand attitude). According to Aaker [13], a brand association is "anything linked in memory to a brand". The relationship between associations and other key concepts, such as brand image, is not always clear [14]. However, Low and Lamb Jr [15] offer some clarity when they conceptualise brand associations as having three dimensions: brand image, brand attitude, and perceived quality.

Human Associative Memory (HAM) theory suggests that humans create associations between different mental elements (such as senses, ideas, data) through experience, that simple ideas will underly these associations, that elementary sensations can be used to identify these simple ideas, and that complex associative configurations can be examined from studying the underlying simple ideas [16]. This theory has been used as the basis for how consumers make brand associations and store them in memory to form brand knowledge, which can be retrieved upon presentation of the brand name [17,18]. This study will discuss HAM in greater detail in the next section, as it uses HAM as the theoretical foundation for the introduction of a new type of brand concept, termed as a compound brand. As mentioned earlier, compound brands are "multi-created". The term "multi-creation" is used to capture the multitude of entities that contribute associations towards compound brands. However, there are several characteristics to this multi-creation that makes compound brands unique:

1.  Compound brands have tenant–landlord relationships with other branded entities;
2.  Compound brands facilitate the value creation of their tenants;
3.  Tenants provide value back to the compound brand through the provision of services to the compound brand's customers;
4.  Compound brands are location-bound and manage an "owned" physical space;
5.  Ancillary entities (e.g., the city, government, transport providers), including non-commercial entities, can also enhance or hinder the value creation processes of the compound brand and its tenants.

In considering the potentially unique characteristics of compound brands and how brand associations are formed in memory according to HAM, this research was premised on the idea that a compound brand's associations will be made up of associations sourced from the compound brand entity itself, from the tenants of the compound brand, and from ancillary entities. All these associations will be linked together in memory and may be recalled upon presentation of the compound brand name. This study also considers it likely that, when retrieving choice sets and making evaluations between brands, these different associations linked in memory will also contribute to brand choice [19,20]. While only airports and shopping malls are studied in this paper, the characteristics outlined above suggest that compound brands may be relatively common in the marketplace, with this paper predicting that the following types of entities may be compound brands:

other transportation hubs (e.g., train stations, bus terminals, ports), hospitals, universities, stadiums, theatres, museums, theme parks, medical centres, casinos, hotels, office buildings, and business parks.

In light of the serendipitous nature of the "finding" of compound brands in the initial study on airports, the research questions were developed post-hoc to be applied to the second case of shopping malls. However, the research questions applied to shopping malls were also applicable to the initial case of airports that prompted the compound brand concept in the first place, and thus the data from the initial study were re-analysed to focus more tightly on investigating the compound brand concept. With these points in mind, this paper aims to answer four research questions, as follows:

1. Do consumer associations support the idea of multi-creation of brand associations to form compound brands?
2. Do compound brand tenants act as sources of brand associations with the compound brand entity itself?
3. Are compound brand tenant associations important in determining choice between compound brands?
4. If the answer to Question 3 is yes, then, are there differences between compound brand entities in terms of the importance of compound brand associations sourced from tenants?

This study commences with a literature review focussed on HAM and establishing clear lines of demarcation between compound brands and potentially related concepts. In line with calls to better blend theory and data [21], the methods section shows how qualitative and quantitative techniques were combined to balance the richness of the data against generalisability and also to show how the analyses relate back to the theory of compound brands. The research findings are then presented, which provide empirical support for the proposed concept of compound brands, and this study proposes that these brands should be considered a new type of brand for both managerial and theoretical purposes. This study ends with a formal definition of a compound brand and presents some thoughts on future research for the concept. In totality, this study contributes to existing knowledge within the marketing literature by identifying a new and distinct type of brand. The distinctive nature of compound brands means that they have theoretical and managerial implications that are distinct from other types of brands, e.g., [22,23]. Further, this study identifies that different compound brands, whilst sharing structural commonalities, also sit on a continuum whereby different compound brands have higher or lower proportions of associations sourced from their tenants and ancillary entities as compared with those sourced from the focal brand (e.g., an airport). This continuum provides a potentially rich vein of future managerial research and will provide impetus for further research to deepen the theoretical understanding of compound brands.

## 2. Literature Review

### 2.1. Human Associative Memory (HAM)

As previously noted, HAM is a theory that suggests humans connect associations together in memory through experience to construct complex associative networks that can be recalled through the presentation of stimuli that link the associations together [16]. Despite the former being commonalities that underpin HAM, there are nevertheless a range of models that provide different accounts of how HAM functions. For example, despite agreement with the premise that humans can relate seemingly unrelated items in episodic memory as part of a common experience, there are different models of how recognition of the associated items occurs. Studies on associative learning and recognition (i.e., how the items become related in memory) tend to fall into three types of models: (1) those that use single-process recall-only models, where cued recall tasks are used to elucidate paired associates and the recall is measured as either a match or a mismatch, e.g., [24,25]; (2) those that use single-process familiarity-only models that assume that recognition of paired associates can be measured by the level of familiarity, which is

viewed as a continuous variable, e.g., [26,27]; and (3) dual-process models, which use both familiarity and recollection details to make judgements as to the speed and accuracy of recognition using likelihood models, e.g., [28,29]. Malmberg and Xu [30] acknowledge the relative strengths in the different models, highlighting that humans are flexible and adopt whichever associative recognition strategy gives the highest accuracy with the greatest efficiency.

One of the underlying assumptions in dual process models is that single-item recognition is the result of a process where a retrieval cue is compared with a large number of memory traces in episodic memory [31,32]. In studies of associative recognition, a common method is to use experiments where participants are given items to study (known as targets) and then must discriminate these from unstudied items (known as foils). Recognition is considered as successful when participants are able to identify targets while rejecting foils [30]. There have been several studies that have highlighted how the similarity of the targets and foils affects recognition accuracy. For example, process discrimination where participants discriminate between items that appeared in similar contexts [33], exemplar discrimination where participants discriminate between exemplars within the same semantic category [34], and plurality discrimination where participants discriminate between singular and plural forms of the same words [35]. While this study does not measure the recognition of compound brand associations (i.e., examining accuracy), the studies on associative recognition show that humans tend to be better at remembering generalities rather than specifics (e.g., are better at discriminating between words with different semantic meanings rather than words with similar semantic meanings) and that the mind uses retrieval cues in working memory as a means of comparing various memory traces present in episodic memory. This study's research method is informed by this, prompting episodic memory by asking for brand associations from recent visits to what are considered potential compound brand entities. These research findings also direct attention to the key role played by episodic memory in the formation of the structure of compound brands.

Episodic memory is the part of long-term memory that stores past experiences [36], whereby associations from experiences are stored and inter-related with each other in memory [16], and where cues in working memory can be compared against memory traces in episodic memory in order for recognition to occur [31]. Using this framework, because consumers have experiences that take place within compound brand entities (e.g., an airport) then such experiences are stored in episodic memory (e.g., each visit to the airport). Therefore, the use of a retrieval cue (e.g., an airport brand name) will trigger the working memory to compare the retrieval cue with the various episodes stored in episodic memory. Because of the presence of tenants and ancillary entities in the same episodes, the single retrieval cue (i.e., the name of the compound brand) will trigger recall of associations sourced from the compound brand entity, its tenants, and ancillary entities related to the different episodes stored in memory. Thus, it can be seen that the "location-bound" nature of compound brands is an important element in making them unique; in other words, the brand associations are derived from the totality of the episode, whereby all the episodic associations (regardless of their source entity) are linked via memory back to the focal brand entity (e.g., an airport).

### 2.2. Similar and Related Concepts

When proposing a new concept, it is important that the concept is clearly delineated from other concepts, avoiding the problems associated with conceptual redundancy [37,38], and establishing that it is a genuinely new concept. In the case of compound brands there are significant overlaps with longstanding brand concepts, such as Leveraged Marketing Communications (LMC), co-branding, and place brands. Because such concepts are well-established, they have large bodies of literature associated with them despite the underlying concepts not changing. With this in mind, it is important to establish the foundations of these concepts and address why the compound brand is a genuinely new concept in relation

to these extant concepts. Therefore, in places, we draw upon older literature to establish fundamental differences between compound brands and the extant concepts in question.

### 2.2.1. Leveraged Marketing Communications (LMC)

LMC can be defined as "brand building strategies that pair a brand with another object for the purpose of enabling the brand to benefit from the associations the target audience has with the object" [39], p. 157. A key premise of LMC is that objects such as people, places, and other entities may be linked to a brand and serve to provide secondary associations with a brand. Keller [40] observes that, for secondary associations to have any impact at all, consumers must first have knowledge about the linked entity (e.g., a person) and the nature of this knowledge needs to be such that consumers are likely to update their knowledge about the brand. Using LMC to build secondary brand associations relies upon actions by brand managers to link their brand to other entities, using techniques such as celebrity endorsement [41]. In some respects, this might be said to apply to compound brands, for example, airports may choose specific restaurants or clothing brands to be present within their facility with a view to enhancing positive brand associations. Although sharing some commonality, the aim of LMC is to create secondary associations and this differs from compound brands. In the case of compound brands, primary associations are formed as consumers view them as part of the product provided by the focal compound brand (e.g., consumers view food facilities as part of the airport's core product). In this sense, rather than the focal compound brand's tenants being seen as indirectly linked to the judged product to form indirect associations [42], their tenants and ancillary entities are providing part of the judged product such that associations sourced from tenants and ancillary entities become primary associations directly linked back to the compound brand name.

### 2.2.2. Ingredient Branding and Co-Branding

There are two existing concepts in the branding literature that may be seen as related to compound brands, but which can nevertheless be conceptually delineated from compound brands. Firstly, ingredient branding typically describes a situation whereby one branded product is used as an ingredient within another branded product (e.g., Intel processors contained within Apple Macs) [43,44]. Desai and Keller [45] identify that ingredient branding combines the existing brand name with a new brand name from the same company or combines the existing brand name with another established brand name. The latter type of ingredient branding is similar to compound brands in that two (or more) different brand names may be closely connected in consumer associations and product evaluations. However, the compound brand concept differs in two ways. Firstly, the brand associations do not just derive from other brands but may also include non-commercial entities such as governmental entities. Secondly, a compound brand requires that all the tenants are located in an "owned" geographic location.

Another similar concept is that of co-branding (also known as a brand alliance), which has traditionally involved two partner brands (also known as constituent brands) that form a new co-brand or "composite" brand [46], whereby the co-brand results from associations being transferred from the constituent brands [47]. However, co-brands may involve multiple partners, the number and diversity of which affects consumer perceptions of the co-brand [48]. Co-brands are similar to the proposed compound brand but differ in that no new brand is being created for a compound brand; instead, the associations transfer from a tenant to the compound brand entity, both of which are usually established brands in their own right.

### 2.2.3. Brand Co-Creation vs. Brand Multi-Creation

The idea of brand co-creation originates from service-dominant logic [49] and considers that the value of brands and brands themselves are co-created by firms and their consumers [50–53]. While this model of brand creation can be widely applied and could be said to be a part of the formation of a compound brand, there are nevertheless unique

characteristics of compound brands which create clear lines of delineation. Although the co-creation may take place at a tenant level, the positive or negative associations from this process will transfer to the focal brand entity and thus serve to "multi-create" the focal brand associations. Thus, the value of the brand is also not created from a single focal brand but is instead derived from multiple brands and ancillary entities. A further delineation is that the transfer of any co-created association may also derive from non-commercial entities (e.g., location). Finally, the source of the associations is derived from actors who have a variable degree of independence (e.g., an airport shop may be semi-independent, but government security is independent).

### 2.2.4. Brand Architecture in Relation to Compound Brands

Brand architecture can be thought of as "the way in which companies organise, manage and go to market with their brands" [22], p. 23. Aaker and Joachimsthaler [23] classify various types of brand architecture strategies into a brand relationship spectrum, with branded house and house of brands as the overarching relationship types. Within the spectrum, the most relevant relationship types to inform an understanding of compound brands are brand endorsement strategies and the house of brands strategy. The latter is described as "an independent set of stand-alone brands" [23], p.10, which captures the fact that compound brands do not share a brand identity. For the former, the strategy involves sharing a brand identity (name or logo) of the parent brand with the endorsed brand, but where the endorsed brand acts independently within the marketplace [54].

Although not always explicit, there is an implicit endorsement of tenant brands by the brand of the focal entity. Compound brands are location-bound, and tenants occupy space within them, meaning the two brand identities will be inadvertently presented next to each other in a similar way to an endorsed brand and its parent brand. Compound brands also share commonalities with houses of brands. The associations of the independent stand-alone brands can be seen as entirely separate in their own right. As such, the individual brands will have the benefits of distinctive associations that accompany this strategy. However, when located within a focal compound brand entity, there are also (either implicit or explicit) endorsement strategies at play. For example, if a less well-known brand of café is in a shopping mall, the café will potentially benefit from the endorsement of the shopping mall brand. Conversely, one might expect that McDonald's is a stronger and more recognisable brand than that of a shopping mall (i.e., the focal brand), meaning that McDonald's is arguably the endorsing brand. Therefore, although endorsement benefits may apply, compound brands differ in the respect that the endorsing relationship may be in either direction.

As the discussion suggests, traditional brand architecture can be applied in part to compound brands and inform understandings of compound brands. However, it should be noted that the focal compound brand can still fit into traditional brand architecture. For example, The Mall of America is owned by the Triple Five Group and is a thus part of a house of brands strategy. As such, compound brands both fit within a traditional brand architecture, and compound brands may share features with some facets of brand architecture, but they nevertheless should be seen as conceptually distinct.

### 2.2.5. Place Brands vs. Compound Brands

Because compound brands occupy a geographic location, a conceptual delineation between place brands and compound brands is needed. For the purposes of this delineation, this paper uses the term "place brand" as an all-encompassing term, including destination brands [55,56], city brands [57], country brands [58], regional brands [59], and any other brand where the branded entity is a geographic location [60,61]. A place brand can be defined as "a network of associations in the consumers mind based on the visual, verbal, and behavioural expression of a place, which is embodied through the aims, communication, values, and the general culture of the place's stakeholders and the overall place design" [62], p. 3. This may appear to capture some elements of compound brands;

however, when examining the literature on place branding, it is apparent that there are some significant differences.

The most fundamental of these differences is that the underlying entity behind the brand is very different. The "place" in a place brand is a geographic location rather than a commercial entity [63]. This difference is important because place brands (as geographic locations) have diffuse control over the entities and stakeholders located within their geographic location [64]. By contrast, compound brands have tightly bound control over which tenants are located within their premises because they are the landlords of these other branded entities. Note here that a governmental entity (e.g., city council) may own a compound brand (e.g., an airport). In this instance, whatever the compound brand is named as becomes the focal brand and the ownership of the compound brand by a non-commercial entity is moot.

Although compound and place brands have differences, there are elements of place branding that can inform the understanding of compound brands. For example, Nghiêm-Phú and Suter [65] highlight that airports and their attributes become associated with place brand names. Similarly, Zenker and Beckmann [66] note that entities such as shopping malls can be important in order for a place to be able to satisfy the needs of certain traveller groups. Because compound brands occupy a physical space and can be quite prominent within their geographic area, they can be important parts of a place brand. Equally, the geographic location of a compound brand could be an important contributor to the value of its physical space (e.g., sufficient population, nearby tourist attractions, etc.). This potentially explains why around three-quarters of all airports are named after the place that they are located in [67]. Similarly, Burns and Warren [68] highlight that the location of regional shopping centres is often the primary discriminator in determining consumer choice. Accordingly, location will likely be part of a compound brand's associations, just as a compound brand may also be part of a place brand's associations.

### 2.2.6. Summary of Commonalities and Differences

Table 1 summarises the core commonalities and differences between compound brands and the extant brand concepts discussed above. Ticks show that a characteristic listed in a row applies to a brand concept listed in the column, while crosses indicate that the characteristic does not apply. One can see clearly that there are some commonalities with each of the extant brand concepts mentioned; however, no extant concept has exactly the same combination of characteristics as compound brands.

**Table 1.** Summary of commonalities and differences between compound brands and other extant brand concepts.

| Characteristic | Leveraged Marketing Communication | Ingredient Branding | Co-Branding | Brand Co-Creation | House of Brands | Brand Endorsement | Place Brands | Compound Brands |
|---|---|---|---|---|---|---|---|---|
| Brand associations sourced from other entities | ✓ | ✓ | ✓ | ✓ | ✗ | ✓ | ✓ | ✓ |
| Primary brand associations sourced from linked entities | ✗ | ✓ | ✓ | ✗ | ✗ | ✓ | ✓ | ✓ |
| Tenant–landlord relationships with linked entities | ✗ | ✗ | ✗ | ✗ | ✗ | ✗ | ✗ | ✓ |
| Focal brand facilitates value creation of linked entities | ✗ | ✓ | ✓ | ✗ | ✗ | ✗ | ✓ | ✓ |
| Linked entities facilitate the value creation of the focal brand | ✓ | ✓ | ✓ | ✗ | ✗ | ✓ | ✓ | ✓ |
| Location bound | ✗ | ✗ | ✗ | ✗ | ✗ | ✗ | ✓ | ✓ |

**Table 1.** *Cont.*

| Characteristic | Leveraged Marketing Communication | Ingredient Branding | Co-Branding | Brand Co-Creation | House of Brands | Brand Endorsement | Place Brands | Compound Brands |
|---|---|---|---|---|---|---|---|---|
| Owned physical space (i.e., control over their space) | ✗ | ✗ | ✗ | ✗ | ✗ | ✗ | ✗ | ✓ |
| Non-commercial entities can enhance or hinder value creation of the focal brand | ✓ | ✗ | ✗ | ✗ | ✗ | ✗ | ✓ | ✓ |
| Co-creation of value (focal brand and consumers) | ✓ | ✓ | ✓ | ✓ | ✓ | ✓ | ✓ | ✓ |
| Multi-creation of value (focal brand, linked entities, and consumers) | ✓ | ✓ | ✓ | ✗ | ✗ | ✓ | ✓ | ✓ |
| Endorsement effects are two-way between the focal brand and linked entities | ✗ | ✓ | ✓ | ✗ | ✗ | ✗ | ✓ | ✓ |
| Endorsement effects may be implicit due to colocation rather than due to an explicit strategy | ✗ | ✗ | ✗ | ✗ | ✗ | ✗ | ✓ | ✓ |
| Each branded entity can be seen as independent and stand-alone | ✓ | ✗ | ✗ | ✗ | ✓ | ✗ | ✓ | ✓ |
| Involves creating a new brand | ✗ | ✗ | ✓ | ✗ | ✗ | ✗ | ✗ | ✗ |
| Endorsement effects flow from a parent brand to an endorsed brand | ✗ | ✗ | ✓ | ✗ | ✗ | ✓ | ✗ | ✗ |

*2.3. Premise for Case Studies: Airports and Shopping Malls*

This study uses the examples of airports and shopping malls as its case studies because these can be considered to be good compound brand candidates. The research aim was to establish whether compound brands are a veridical concept by answering the four research questions highlighted earlier. This study chose airports and shopping malls as the case studies partly because they meet each of the five unique characteristics of compound brands, but also for pragmatic reasons: both entities are used by most consumers with some degree of regularity. Accordingly, one can expect that most consumers are able to recall associations with airports and shopping malls. As highlighted in the literature review, it can also be reasonably expected that at least some of the branded tenants of these entities are better known than the branded entities themselves (i.e., Lacoste may be better-known than Soekarno-Hatta International Airport, and Starbucks may be better-known than the Dubai Mall). Accordingly, if the compound brand concept is veridical, then this should become apparent in the research.

**3. Methods**

*3.1. Participants*

3.1.1. Airports

Out of the 240 participants interviewed about airports, 43.75% were male and 56.25% were female. New Zealand residents (including dual citizens) made up 66.25% of the sample, with 33.75% of the sample representing overseas visitors. The average age of participants was 39.18 ($SD$ = 17.11, range 16–83 years old). A total of 73.33% of participants were employed or self-employed, 4.16% unemployed, 13.75% students, 5% retired, and 3.75% full-time parents. The interviews covered 642 airport visits, including 88 airports spread across 36 different countries. This was possible because every trip using air travel

will involve at least two airports, with some participants transiting through multiple airports as part of their trip.

In terms of the recency of the airport visits, 26.67% of participants had travelled through the airports in the last fortnight, 31.25% within the last three months, 24.58% within the last year, 13.75% within the last three years, and 3.75% over three years ago. Out of the airport visits discussed, 21.5% were the first time the participant had visited that airport, 12.46% had been visited 1–2 times prior, 15.26% 3–5 times prior, 8.57% 6–10 times prior, 30.06% 10–50 times prior, and 12.15% had been visited more than 50 times prior. The primary purpose of the travel also varied among participants, with 35.42% travelling to visit friends and relatives, 32.92% travelling for a holiday or for leisure, 16.25% travelling for business, 3.33% travelling for education, and 12.08% travelling for other reasons.

3.1.2. Shopping Malls

Out of the 240 participants interviewed about shopping malls, 48.33% were male, 51.25% were female, and 0.42% were non-binary. New Zealand residents (including dual citizens) made up 76.67% of the sample, with 23.33% of the sample representing overseas visitors. The average age of participants was 45.53 (*SD* = 19.04, range 16–86 years old). A total of 61.25% of the participants were employed or self-employed, 8.33% unemployed, 12.5% students, 17.08% retired, and 0.83% full-time parents.

The interviews covered 240 different individuals' shopping mall visits, including 35 shopping malls spread across seven countries. Of the most recent shopping mall visits of participants (i.e., the one they were interviewed about), 32.5% of them were on the same day as the interview, 39.58% were within the last week, 12.92% within the last month, 8.33% within the last three months, and 6.25% were more than three months ago. The number of past visits participants had made to the shopping mall they last visited also varied: 19.17% had visited less than five times, 18.33% had visited 5–50 times, 12.5% had visited 50–100 times, and 48.33% had visited more than 100 times. Regarding the purpose of their most recent shopping mall visit, 54.17% of participants were shopping for something specific, 11.67% were there for food or drink (other than groceries), 11.25% were there to visit a different type of tenant (i.e., not a shop or food provider), 7.5% were there to go shopping as an activity, 5.42% were there to spend time with friends or family, 4.58% were there to have a walk or look around, 3.75% were there to fill in time, and 1.67% were there for other purposes.

*3.2. Materials*

We used semi-structured interviews to collect data from participants. Two different instruments were used for the semi-structured interviews, one for airports (see Appendix A) and one for shopping malls (see Appendix B). It should be noted that the interviews relied on unprompted recall of airport and shopping mall brand names, and also unprompted recall of brand associations and participant views of what were important brand associations for choosing between airports or shopping malls. This methodology was premised on two very important principles. The first was to ensure that the brand name was used as the retrieval cue to capture the complex network of brand associations connected to it in episodic memory (as per HAM), thus ensuring that the associations captured were de facto primary brand associations because they linked directly back to the airport brand in question. The second was to avoid self-generated validity, in other words, to avoid creating associations that did not already exist in long-term memory by including measures that assume the sorts of associations participants might already have [69,70]. The approach also aligned with recent calls for more open-ended free association questions when studying brand associations [71]. As a probe for episodic memory and to ensure ease of answering the interview questions, participants were asked about the airports they travelled through on their most recent trip using air travel (for the airport study) or the shopping mall that they most recently visited (for the shopping mall study).

Each semi-structured interview format was piloted, the first with fifteen participants, and the second with five participants. Minor changes to wording were suggested and incorporated, such as the alternative wording for questions presented in Appendices A and B.

### 3.3. Procedure

Convenience sampling was used, with the first author completing street intercepts whilst standing on major thoroughfares in Palmerston North and Wellington in the Lower North Island of New Zealand. For Palmerston North, this was in the central city in the vicinity of Te Marae o Hine—The Square. For Wellington, this was down Cuba Street, a road closed to motor vehicles that has good flow of pedestrians (including tourists). No interviews were completed at an airport (for the airport study) or shopping mall (for the shopping mall study) as that would mean that only visits to those specific airports or shopping malls would be recalled. Both cities have airports with scheduled airline flights, and both cities have shopping malls. Participants were presented with an information sheet outlining what the study was about and the recruitment criteria. Participants needed to be at least 16 years old, to have been to an airport or shopping mall before, and not be employed within an airport or shopping mall. If participants consented to be interviewed and met the recruitment criteria, then they were interviewed in situ by the first author. The two interview formats were administered independently of each other (i.e., no participant did both). This was to ensure that participants did not confuse the two entities (due to the similarity of the questions) and also to avoid participant fatigue or other such order effects [72]. The interviews were recorded on a smart phone or tablet and then later transcribed. Due to the exploratory nature of this research, there needed to be enough data to estimate the veridicality of the concept. Based upon initial interviews, 240 participants for each entity were estimated to be a reasonable and pragmatic number, for a total of 480 participants. Both studies were peer-reviewed and deemed to be low-risk and were therefore registered as such on the Massey University Human Ethics Database.

### 3.4. Analysis

For the purposes of the analysis and answering the research questions, this study provides definitions for two key terms:

1. *Associations* were anything consumers linked in memory to the airport or shopping mall brand name;
2. *Important associations* were associations that consumers used to choose between different airports or shopping malls.

Note that a few associations (less than 50 collectively) were removed because they were generic and not related to a specific branded entity of an airport or shopping mall (e.g., being excited to travel applies to any airport and enjoying shopping applies to any shopping mall). Associations and important associations were grouped using thematic analysis following Braun and Clarke's 15-point checklist [73]. While all the associations and important associations were made with an airport in the minds of the participants, if a participant mentioned a food provider, shop, or location then the association or important association was grouped to the corresponding entity (i.e., tenants for food providers and shops and ancillary entities for location). These groupings were reported with descriptive data.

In line with other qualitative brand research, this study also used quantitative means of analysis to better understand the relationships between the themes and how they contributed towards brand association structures and brand choice, e.g., [74–76]. The combination of approaches recognised the importance of avoiding the extremes of marketing research where the research either only provides rich descriptions of behaviours without generalisability or where data are mined and analysed without prior thought about what might be found and how the data might be explained [21]. Specifically, One-Sample Wilcoxon Signed Rank tests [77] were used to test whether associations sourced from tenants and ancillary entities were statistically significant contributors to airport's association structures. The median percentages of associations and important associations sourced

from tenants and ancillary entities were tested against 0 to see whether the proportion of associations from these entities was significantly different from 0. A significant result meant that these entities contributed to the overall brand association structure of airports.

## 4. Results and Discussions

### 4.1. Airports

When participants were asked to identify the airports that they travelled through on their most recent trip, 87.38% of airports were correctly identified with their brand name (or very close to it: words such as "international" were not deemed consequential). However, 12.62% of airports mentioned were of the location only (i.e., where the participant mentioned the location of the airport but did not know its official name). Correct brand names were taken from the airport's website as some airports are branded under more than one name (e.g., Los Angeles International Airport is also branded as LAX).

Across all airport visits, participants made 2049 associations, 1303 of which appeared to be unique. The median number of associations was three per airport visit (*IQR* = 1–4). For 3.73% of airport visits, participants made associations with the airport by mentioning the brand name of one of its tenants. These included airlines (e.g., British Airways), food providers (e.g., Subway), bookshops (e.g., Relay), and other tenants. It is also worth pointing out that 18.85% of airport visits resulted in no associations with the airport itself (i.e., the focal branded entity), but only 7.17% of airport visits resulted in no associations at all.

Between the participants, there were 896 important associations, 605 of which appeared to be unique. The median number of important associations was three per participant (*IQR* = 2–5). For important associations, only one participant mentioned a specific brand of tenant. A total of 19.58% of participants had no important associations with the airport itself (i.e., the branded entity), compared with only 5% of participants that had no important associations for choosing between airports. An overview of the associations and important associations made with entities at airports is shown in Table 2.

With regard to airports, Table 2 shows that participants tended to associate airport brands with tenants such as airlines, food and beverage providers, and shops, among others. There were also ancillary entities such as the city that the airport was located in and the government-imposed security measures. According to the answers of participants, all of these were connected by the brand name of the airport. Demonstrably, for airports, the answer to Research Question 1 is "yes"—consumers do multi-create brand associations to create compound brands.

The descriptive data shown in Table 2 support the idea that tenants provide a source of brand associations for airports (i.e., the answer to Research Question 2 is yes) because 14.59% of all brand associations with airports were sourced from tenants, and 28.97% of all airport visits had at least one association sourced from tenants. If tenants did not act as a source of brand associations for airports, then one would expect that the median percentage of brand associations sourced from tenants would not be statistically significantly different from 0% across participants. A One-Sample Wilcoxon Signed Rank Test revealed a statistically significant difference between the observed median of 7.69% and the hypothetical median of 0%, $z = 9.744$, $p < 0.001$, with a large effect size ($r = 0.64$) (please see explanatory note 1 in Appendix C). This provides empirical evidence in support of Research Question 2. The same procedure can be followed for examining Research Question 3—that is whether or not tenants act as a source of important associations for choosing between compound brand entities (in this case airports). A One-Sample Wilcoxon Signed Rank Test revealed a statistically significant difference between the observed median of 20.71% and the hypothetical median of 0%, $z = 9.880$, $p < 0.001$, with a large effect size ($r = 0.65$). In addition, Table 2 shows that 26.34% of important associations were sourced from tenants and 53.75% of all participants had an important association sourced from tenants. These results provide empirical support for an affirmative answer to Research Question 3.

To add further support to Research Question 1, One-Sample Wilcoxon Signed Rank Tests were also run to see whether the percentages of associations and important associations sourced from ancillary entities were statistically significant. For associations, the test revealed a statistically significant difference between the observed median of 16.67% and the hypothetical median of 0%, z = 11.077, $p < 0.001$, with a large effect size (r = 0.72). For important associations, the test revealed a statistically significant difference between the observed median of 0% and the hypothetical median of 0%, z = 8.311, $p < 0.001$, with a large effect size (r = 0.54) (please see explanatory note 2 in Appendix C).

**Table 2.** Associations and important associations sourced from entities at airports.

| Entity | Associations | | Important Associations | | Examples [4] |
|---|---|---|---|---|---|
| | Percentage [1] | Percentage of Visits [2] | Percentage [1] | Percentage of Participants [3] | |
| Airport | 66.57% | 81.15% | 59.82% | 80.42% | |
| Airport | 64.52% | 80.53% | 57.37% | 79.17% | Facilities, atmosphere, design, airport service quality |
| Transport (within airport control) | 2.05% | 5.92% | 2.46% | 7.92% | Parking, buses between terminals |
| Tenants | 14.59% | 28.97% | 26.34% | 53.75% | |
| General | 1.95% | 5.92% | 3.24% | 11.67% | Variety of services available |
| Airlines | 4.29% | 10.90% | 7.37% | 19.58% | Check-in procedures, airline staff, airline brand names |
| Food and Beverage | 4.29% | 11.06% | 9.49% | 27.92% | Restaurants, cafés, bars, types of cuisine |
| Shops | 3.76% | 9.81% | 5.13% | 14.17% | Duty free, clothing, cosmetics, bookstores, souvenir shops |
| Others | 0.29% | 0.93% | 1.12% | 3.75% | Hotels, banks, phone companies, rental car companies |
| Ancillary Entities | 18.84% | 39.56% | 13.84% | 38.75% | |
| Customers | 3.56% | 9.19% | 1.90% | 6.25% | User imagery, number of people |
| Government | 3.90% | 8.72% | 5.8% | 18.33% | Security, customs, immigration |
| Location | 9.32% | 22.12% | 2.23% | 8.33% | City, country, views, weather, local attractions |
| Transport (outside of airport control) | 2.05% | 5.3% | 3.91% | 12.08% | Buses, trains, taxis, roads |

[1] Percentages of associations and important associations were calculated by dividing the number in each category by the total number. [2] Percentage of visits was calculated by dividing the number of visits with at least one association with the entity by the total number of visits. [3] Percentage of participants was calculated by dividing the number of participants with at least one important association by the total number of participants. [4] The examples column is not exhaustive and only presents a few prominent examples for each entity.

### 4.2. Shopping Malls

When participants were asked to identify the shopping mall they most recently visited, 76.67% of shopping malls were correctly identified with their brand name (or very close to it: differences between words such as "mall" and "centre" were not deemed consequential) and 23.33% of shopping malls mentioned were of the location only (i.e., where the participant mentioned the location of the shopping mall but did not know its name). Correct brand names were taken from the shopping malls' websites as some shopping malls are branded under more than one name.

Across the shopping mall visits, participants made 773 associations, 476 of which appeared to be unique. The median number of associations was three per shopping mall visit (*IQR* = 2–4). For 16.67% of shopping mall visits, participants made associations with the shopping mall by mentioning the brand name of one of its tenants. These were primarily food providers (e.g., McDonald's) and retail stores (e.g., Kmart), but also included supermarkets, banks, technology stores, and phone companies, among others. Interestingly,

54.17% of shopping mall visits resulted in no associations with the branded entity itself (i.e., the shopping mall), but only 2.5% of shopping mall visits resulted in no associations at all.

Participants had 679 important associations, 424 of which appeared to be unique. The median number of important associations was two per participant (*IQR* = 1–4). For important associations, 6.67% of participants mentioned the brand name of a tenant. These were all brands of shops (e.g., Cotton On, Kmart, Nike). A total of 40.83% of participants had no important associations with the branded entity itself, compared with only 4.17% of participants who had no important associations for choosing between shopping malls.

With regard to shopping malls, Table 3 shows that participants created associations with shopping mall brands that were sourced from tenants and ancillary entities. According to the responses of the participants, all of these were connected by the brand name of the shopping malls, providing further support in favour of Research Question 1—consumers do multi-create brand associations to create compound brands.

**Table 3.** Associations and important associations sourced from entities at shopping malls.

| Entity | Associations | | Important Associations | | Examples [4] |
|---|---|---|---|---|---|
| | Percentage [1] | Percentage of Visits [2] | Percentage [1] | Percentage of Participants [3] | |
| Shopping Mall | 30.14% | 45.83% | 37.56% | 59.17% | |
| Shopping Mall | 28.59% | 40.83% | 30.93% | 42.5% | Facilities, atmosphere, design |
| Transport (within shopping mall control) | 1.55% | 5.00% | 6.63% | 16.67% | Parking |
| Tenants | 58.99% | 77.92% | 54.34% | 74.58% | |
| General | 7.76% | 17.5% | 10.31% | 25.00% | Variety of services available, price point |
| Food and Beverage | 13.20% | 32.92% | 10.16% | 22.92% | Restaurants, cafés, food courts, fast food outlets, grocery stores, bars |
| Shops | 34.54% | 59.17% | 32.11% | 53.33% | Retail, clothing, technology, bookstores, variety of shops |
| Others | 3.49% | 8.33% | 1.77% | 4.58% | Banks, phone companies, optometrists |
| Ancillary Entities | 10.87% | 26.67% | 8.10% | 20.83% | |
| Customers | 7.50% | 17.92% | 1.47% | 4.17% | Number of people, customer behaviour, user imagery |
| Location | 2.98% | 7.50% | 3.53% | 9.58% | City, proximity to other places |
| Transport (outside of airport control) | 0.39% | 1.25% | 3.09% | 7.08% | Public transport, roads |

[1] Percentages of associations and important associations were calculated by dividing the number in each category by the total number. [2] Percentage of visits was calculated by dividing the number of visits with at least one association with the entity by the total number of visits. [3] Percentage of participants was calculated by dividing the number of participants with at least one important association by the total number of participants. [4] The examples column is not exhaustive and only presents a few prominent examples for each entity.

The descriptive data shown in Table 3 support the idea that tenants provide a source of brand associations for shopping malls (i.e., the answer to Research Question 2 is yes) because 58.99% of all brand associations with shopping malls were sourced from tenants, and 77.92% of all shopping mall visits had at least one association sourced from tenants.

A One-Sample Wilcoxon Signed Rank Test revealed a statistically significant difference between the observed median percentage of tenant associations of 75% and the hypothetical median of 0%, z = 12.19, *p* < 0.001, with a large effect size (r = 0.80). This provides empirical evidence in support of Research Question 2. The same procedure can be followed for examining Research Question 3 (tenants as sources of important associations). For important associations, a One-Sample Wilcoxon Signed Rank Test also revealed a statistically significant difference between the observed median of 66.67% and the hypothetical median of 0%, z = 11.808, *p* < 0.001, with a large effect size (r = 0.78). In addition, Table 3 shows that 54.34% of important associations were sourced from tenants and 74.58% of all participants had an important association sourced from tenants. These results provide empirical support for an affirmative answer to Research Question 3.

To add further support to Research Question 1, One-Sample Wilcoxon Signed Rank Tests were also run to see whether the percentages of associations and important associations sourced from ancillary entities were statistically significant. For associations, the test revealed a statistically significant difference between the observed median of 0% and the hypothetical median of 0%, z = 6.634, *p* < 0.001, with a medium effect size (r = 0.43) (please see explanatory note 3 in Appendix C). For important associations, the test revealed a statistically significant difference between the observed median of 0% and the hypothetical median of 0%, z = 5.533, *p* < 0.001, with a medium effect size (r = 0.36) (please see explanatory note 3 in Appendix C).

*4.3. Comparison between Airports and Shopping Malls*

Wilcoxon Signed Rank Tests were performed to see whether the difference between the median percentage of associations and important associations sourced from tenants were statistically significantly different between airports and shopping malls. For associations, the median for shopping malls of 75% was statistically significantly higher than the median of 7.69% for airports, z = 12.311, *p* < 0.001, with a large effect size (r = 0.80). A similar result was obtained for important associations, where the median for shopping malls of 66.67% was statistically significantly higher than the median of 20.71% for airports, z = 10.015, *p* < 0.001, with a large effect size (r = 0.66). These results support an affirmative answer to Research Question 4 (that the importance of associations sourced from tenants varies according to the compound brand entity).

*4.4. Brand Association Multi-Creation*

The proportionality of brand associations for airports and shopping malls supports the idea that the unique characteristics of compound brands result in the multi-creation of brand associations between different entities because both tenants and ancillary entities were the source of a statistically significant percentage of brand associations, and these can be described as primary associations. The combination of the different entities within the focal brand leads to associations from those entities becoming connected with the brand name of the compound brand. In some cases, participants were unable to correctly recall the brand name of the focal compound brand (12.62% for airports and 23.33% for shopping malls). The finding that so many participants could not even correctly name the compound brand supports the idea that episodic memory about compound brands was stronger than the brand name (i.e., sematic memory) for many participants.

The structure of the brand associations and important brand associations also highlights the interconnectedness of associations related to different entities. Tenants comprised a significant part of the association structure for both airports and shopping malls (14.59% for airports and 58.99% for shopping malls). This is in line with the theorised model of the unique characteristics of compound brands. The same can be said about tenants' contributions to the important associations that were used to choose between compound brands (26.34% for airports, 54.34% for shopping malls). For many participants (19.58% for airports, 40.83% for shopping malls), the branded entity itself (i.e., the airport or shopping mall) did not feature in the important brand associations used to choose between airports and

shopping malls. Accordingly, those participants would choose between airports and shopping malls solely based upon their tenants and ancillary entities. These findings combined with the statistical analyses confirm that tenants not only comprise a significant portion of the compound brand's association structure, but also that the associations sourced from the influence of the tenant's choice in relation to compound brands. This was also true for ancillary entities as demonstrated with the statistical analyses, supporting the importance of the fifth unique characteristic of compound brands. However, this paper places limited emphasis on these statistics as the focal brand entity often has limited or no control over the operations of ancillary entities.

### 4.5. The Compound Brand Continuum

As can be seen when comparing between Tables 2 and 3, while the underlying association structures for airports and shopping malls were sourced from the same sorts of entities, their size and importance varied. It seems that airports play a more significant role than their tenants in creating brand associations and in determining brand choice (e.g., airports have some control over flight connectivity and service standards). However, for shopping malls, tenants create more of the brand associations and have more influence over brand choice than the shopping mall itself as the branded entity. The results of the Wilcoxon Signed-Rank Tests also confirm that the differences between airports and shopping malls in this regard were statistically significant with a large effect size, where tenants were much bigger contributors towards brand associations and important brand associations for shopping malls than for airports.

These results relate back to the functioning of episodic memory: associations become connected through experience, so it is a participant's past experience that determine the structure of brand associations in memory [17,78]. Importantly, the differences in the proportionality of associations and important associations between airports and shopping malls suggests that a continuum exists as to how important tenants are as a source of brand associations and in determining brand choice. This paper would expect that other potential compound brands will sit in different locations along this continuum despite having the same underlying sources of brand associations (i.e., focal branded entity, tenants, and ancillary entities). For example, many hotels also have tenants (such as food outlets and convenience stores); however, they are likely to play less of a role in hotel guest experiences than tenants do in air traveller experiences at airports.

### 4.6. Defining Compound Brands

This paper considered that there are certain characteristics that delineate compound brands from other brand concepts. In addition to the review, the findings of the research support the introduction of compound brands as a distinct concept. In line with many theorists, this paper considers it important to provide a clear definition when introducing a new concept and, importantly, a definition which sets clear boundaries around the concept, e.g., [79–81]. This paper therefore provides a definition for compound brands that is unambiguous, but also with clear boundaries to delineate the concept from extant concepts.

The definition is: A compound brand is a focal branded entity whereby its brand associations are multi-created with associations sourced from other entities such that these associations become part of the focal brand's associations. Specifically, they must include "tenant" associations, but may also include associations from ancillary entities such as customer associations, location associations, and transport associations. In order for a brand to be described as a compound brand, it requires that there are tenants within its "owned" physical space that can contribute to the compound brand's associations, and that the compound brand facilitates the value creation of the tenants and vice versa.

There are three key elements of this definition that are worthy of further elaboration. Firstly, a compound brand represents an "owned" geographic space where tenants and ancillary entities are co-located. As discussed, this is not the same as a place brand. Secondly, tenants and ancillary entities are a source of associations from which the compound brand

is comprised. Thirdly, there must be tenants such that the distinct relationship between tenants and the compound brand (as landlord) makes it fundamentally different from relationships such as co-branding. These points in the definition mean that a compound brand does not overlap with an entity such as a store which stocks multiple brands.

*4.7. Managerial Implications*

Many of the associations found for airports and shopping malls are associations that might be expected by managers of those entities. However, extant research tends to focus on very specific types of associations and sources of associations, such as aspects of the physical environment or atmosphere, types of facilities/products on offer, and so on [82–84]. While some have been more holistic, their focus has not been on the structure of brand associations, e.g., [65]. While this paper reiterates many of the findings of the extant literature, it puts those findings within a framework that helps to explain why those things are important and to understand their impact upon consumer brand associations formed in episodic memory, thus providing a more holistic view for managers. In doing so, managers can now assign importance to different sources of associations, which then determines the level of management time and attention paid to those sources. The compound brand continuum is helpful in this regard for identifying general differences between the proportion of brand associations sourced from tenants based upon the compound brand entity in question (e.g., shopping mall managers must always see tenants as a critical part of their brand strategy, but hotels will likely see tenants as a lesser priority). However, there will also be differences between specific entities (e.g., a small airport may have only one shop, whereas a larger one may have over 100) that will determine levels of importance. Regardless, as tenants can be seen to influence compound brand choice, it is of vital importance for managers to understand what factors consumers are using to determine brand choice and the relative importance of those factors [85,86], taking into account the unique situation of their compound brand entity.

Another important contribution is to highlight that ancillary entities can also contribute primary brand associations towards the compound brand. Managers may have limited control over these entities, but they cannot be ignored because of their importance. For example, airport security is found to be a major contributor towards airport brand associations. While security policy is decided at a governmental level and airports will need to comply with the relevant legislation, airport managers can still influence brand associations through strategies such as allocating spaces that allow more checkpoint lanes or investing in technologies that could expedite the flow of passengers through security [87–89]. The importance of managing the impact of these ancillary entities will again depend upon the type of compound brand (e.g., airport or shopping mall) as well as the specifics of the entity in question (e.g., small vs. large, simple vs. complex).

**5. Conclusions**

This paper introduced the concept of compound brands to the marketing literature, providing a theoretical foundation and supporting its validity through empirical research. Compound brands, such as airports and shopping malls, often have brand associations that come from the tenants within them. This is a significant finding with both theoretical and managerial implications. There has been a lot of interest in how brands are constructed and stored in memory, and how brand associations influence consumer choice. This research has previously focussed on various types of brands, including place brands and product brands. However, compound brands represent a new category of brand that is likely to be prevalent in the marketplace. Understanding compound brands can improve our understanding of consumer choice. This research specifically examined airports and shopping malls as examples of compound brands. Future research will identify more categories of compound brands, which may vary on the compound brand continuum but share common underlying characteristics. It will be valuable for theorists, researchers, and managers to understand the positioning of different types of compound brands on this continuum.

## 6. Limitations and Future Research

This paper focussed on whether compound brands are veridical. However, further research is needed to fully understand the managerial implications of the unique structure of brand associations in compound brands. The paper also examined airports and shopping malls as general examples of compound brands, but more specific case studies may be useful for practitioners.

While this paper only explored two types of compound brands (airports and shopping malls), it is likely that other types exist. Examples may include transportation hubs (e.g., train stations, bus terminals, ports), hospitals, universities, stadiums, theatres, museums, theme parks, medical centres, casinos, hotels, and business parks. Future research is needed to determine whether these are indeed examples of compound brands and to understand their position on the compound brand continuum in terms of the role of tenants in their brand association structures.

Another interesting avenue for future research would be to compare the relationships between tenants and compound brands with the relationship between principals and agents. This relationship, often studied in the context of executive compensation, acknowledges that executives (as agents) have control over a corporation, but are not owners [90,91]. They must therefore act in the best interests of the shareholders (the principals) who own the corporation. There is a similar dynamic in the relationship between tenants and compound brands, as tenants have some control over the brand but are not owners. Further exploration of this topic through the lens of principal–agent theory could be valuable.

**Author Contributions:** Conceptualisation, I.L.H. and M.A.; formal analysis, I.L.H.; investigation, I.L.H.; methodology, I.L.H., M.A., W.H.K.T., T.N. and A.G.; supervision, W.H.K.T., M.A. and A.G.; validation, M.A.; writing—original draft, I.L.H.; writing—review and editing, I.L.H., M.A., W.H.K.T., T.N. and A.G. All authors have read and agreed to the published version of the manuscript.

**Funding:** This research received no funding.

**Institutional Review Board Statement:** The two studies that comprise the paper (i.e., one on airports, and one on shopping malls) were peer-reviewed and deemed to be low-risk. They were registered as such on the Massey University Human Ethics Database.

**Informed Consent Statement:** Informed consent was obtained from all subjects involved in the study.

**Data Availability Statement:** Transcripts of interviews can be made available upon written request to the corresponding author.

**Conflicts of Interest:** The authors declare no conflict of interest.

## Appendix A. Interview Questions for Airports

1. Could you please state your:
   - Gender
   - Age
   - Occupation
   - Nationality
2. How often do fly?
3. Think of the most recent time you flew somewhere.
4. When was it?
5. What was the purpose of the trip?
6. Which airport did you depart from?
7. How long did you spend at that airport?
8. Was that your first time travelling through that airport? (If not, how many times have you previously travelled through that airport?)
9. Which airline were you flying on?
10. Which class were you flying in?
11. How long was the flight?

12. Which airport did you arrive at next?
13. Was this for transit, or what it your destination?
14. How long did you spend at that airport?
15. Was that your first time travelling through that airport? (If not, how many times have you previously travelled through that airport?)
16. (If transiting, go back to question 9)
17. Continue until all airports are covered.
18. Was there a return flight?
19. Did you return home using the same route? (if not, then cover other airports too)
20. Thinking back to the airport you departed from when you began your trip, what associations do you make with that airport? (If participants do not understand, this can be rephrased to: "What comes to mind when I say (airport name)?")
21. Think back to the next airport you went through on that trip, what associations do you make with that airport? (If participants do not understand, this can be rephrased to: "What comes to mind when I say (airport name)?")
22. Continue until all airports are covered.
23. If you were given a choice between airports, which associations would be important in making your decision? (If participants do not understand, this can be rephrased to: "If you imagine that you are in a situation where you can choose between several airports to travel through, what sort of things would be important in choosing which one you would rather go through?")
24. Why are those things important?
25. Any further comments?

## Appendix B. Interview Questions for Shopping Malls

1. Could you please state your:
   - Gender
   - Age
   - Occupation
   - Nationality

2. How often do you visit shopping malls?
3. How long do you typically spend at shopping malls at each visit?
4. Think of the most recent time you visited at a shopping mall.
5. When was it?
6. What was the purpose of the visit?
7. Which shopping mall was it?
8. How long did you spend at that shopping mall?
9. How many times had you been to that shopping mall before?
10. Thinking about the last shopping mall that you visited, what associations do you make with that shopping mall? (If participants do not understand, this can be rephrased to: "What comes to mind when I say (shopping mall name)?")
11. If you were given a choice between shopping malls, which associations would be important in making your decision? (If participants do not understand, this can be rephrased to: "If you imagine that you are in a situation where you can choose between several shopping malls to go to, what sort of things would be important in choosing which one you would rather visit?")
12. Why are those things important?
13. Any further comments?

## Appendix C. Explanatory Notes for Statistical Analyses

Explanatory Note 1: The figures of percentages were used for calculating the differences between medians in the Wilcoxon Signed Rank Tests because the proportion of associations sourced from different entities might be a more useful measure than the raw

number of associations. This is likely given the large differences between participants in the number of associations made.

Explanatory Note 2: The figure of 0% was the observed median because more than 50% of participants had 0% of their important associations sourced from ancillary entities. The observed mean was 19.94%, but it would be inappropriate to use the mean for statistical tests due to the skewness of the data.

Explanatory Note 3: 0% was the observed median because more than 50% of participants had 0% of their associations and important associations sourced from ancillary entities. The means for associations and important associations sourced from ancillary entities were 9.81% and 8.48%, respectively. However, it would be inappropriate to use the means for statistical analyses due to the skewness of the data.

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
