# Peer review of "Compound Brands and the Multi-Creation of Brand Associations: Evidence from Airports and Shopping Malls"

_sustainability, doi:10.3390/su15021450_

Round 1
Reviewer 1 Report
This paper is an interesting study analyzed through empirical data related to compound brands. Also, the data and papers are well organized. However, in the literature review, it would like to be corrected and supplemented according to the research topic instead of writing a subheading with “2.1. Airports”. In other words, “2.1. Airports” is data to investigate how a compound brand is performing. Therefore, the subheadings of the theoretical background needs to be modified. This paper is an interesting study analyzed through empirical data related to compound brands. Also, the data and papers are well organized. However, in the literature review, it would like to be corrected and supplemented according to the research topic instead of writing a subheading with airport. In other words, “2.1. Airports” is data to investigate how a compound brand is performing.
Author Response
We thank the reviewer for their time spent reviewing our manuscript. Please see our response to the comments in the attached document.

Reviewer 2 Report
Compound brands and multi-creation of brand associations: 2 Evidence from airports and shopping malls
Dear authors,
It is a pleasure to assess your essay. Below, please find my comments and recommendations:
Abstract
- Please include 2-3 quantitative achievements.
- In addition, in the context of your topic, you should provide the research objectives and problems.
Introduction
- In line 24, you mention a new marketing concept. I read the entire paragraph and do not see which is that new marketing concept. Can you explain? Is it LMC? If so, you have to rewrite the paragraph for better understanding.
- In line 34, a new paragraph is initiated, and you start with “The initial investigation….” Which investigation? Yours? Who conducted this investigation?
Literature review
- The introduction should reflect what the paper will be in terms of variables. I did not see in your introduction anything about Episodic Memory. Also, in the introduction, you talk about the brand association, co-brands, and compound brands, and you do not support those variables with literature.
- You present a literature review for airports, but you do not present a literature review for shopping malls. Why?
- This section MUST be reviewed because it is challenging what the authors are trying to validate in terms of literature.
- Do you have hypotheses or theorems that support this study? What do you want to validate?
- Can the authors provide readers with a more comprehensive understanding of the literature review that supports this paper, “state of the art”?
- This section must be integrative and improved significantly. In addition, all variables that are part of the analysis and conclusions must be supported.
Study 1 Methods
- Can the authors explain how they obtained permission and how the participants were interviewed?
- How many days did this process last? Where? Right out of customs? Where?
- Who conducted this interception survey?
- Can the authors explain how they obtained permission for this interception survey? Who gave the permission?
- The interview process covered 642 airport visits, including 88 airports and 36 countries. Can you explain that?
- Did the research team design the questionnaire? If so, did you validate the questionnaire?
- If you used existing scales, can you let the readers know who created those scales?
- Can you cite the names of the cities in which these interviews were conducted?
- Why convenience sample? Can you explain?
Results
- You have enough information to present quantitative results. Regression analysis should be performed to validate the associations between those variables.
- If you used One Sample Wilcoxon Signed Rank Test, you assumed the data is not normally distributed. Can you explain?
- You used terms such as brand architecture, co-branding, and others, but how could those be measured in those interviews and questionnaires?
- I have the same questions about the methodology for shopping malls, mainly because you say you used the same sampling process, etc.
Discussion
- I cannot appreciate a discussion, especially if you are trying to present a theoretical concept. The only theoretical relationship you have used in your discussion section is with references 74 and 75.
- This paper could be interpreted as self-reference, which is totally negative if you want to introduce a new concept.
Conclusions must be improved.
Limitations must be improved.
Finally, I wish you the best in this peer-review process,
Regards,
Author Response
We thank the reviewer for their time spent reviewing our manuscript. Please find responses to all comments in the attached document.

Round 2
Reviewer 2 Report
Dear Authors,
Thank you very much for improving the original paper based on reviewers' observations. I consider the current version has been sufficiently improved to warrant publication in Sustainability.
Good luck!